# Emergence of a novel chikungunya virus strain bearing the E1:V80A substitution, out of the Mombasa, Kenya 2017-2018 outbreak

**Fredrick Eyase**[1,2,3]*, **Solomon Langat**[1], **Irina Maljkovic Berry**[4], **Francis Mulwa**[1], **Albert Nyunja**[1], **James Mutisya**[1], **Samuel Owaka**[1], **Samson Limbaso**[1,2], **Victor Ofula**[1], **Hellen Koka**[1], **Edith Koskei**[1], **Joel Lutomiah**[1,2], **Richard G. Jarman**[4], **Rosemary Sang**[1,2]

1 Department of Emerging Infectious Diseases, United States Army Medical Research Directorate-Africa, Nairobi, Kenya, 2 Center for Virus Research-Kenya Medical Research Institute, Nairobi, Kenya, 3 Institute for Biotechnology Research-Jomo Kenyatta University of Agriculture and Technology, Juja, Kenya, 4 Walter Reed Army Institute of Research, Silver Spring, Maryland, United States of America

* fredrickeyase2012@gmail.com

**Data Availability Statement:** All relevant data are within the manuscript. Sequence reads are deposited in SRA accession number:

## Abstract

Between late 2017 and mid-2018, a chikungunya fever outbreak occurred in Mombasa, Kenya that followed an earlier outbreak in mid-2016 in Mandera County on the border with Somalia. Using targeted Next Generation Sequencing, we obtained genomes from clinical samples collected during the 2017/2018 Mombasa outbreak. We compared data from the 2016 Mandera outbreak with the 2017/2018 Mombasa outbreak, and found that both had the *Aedes aegypti* adapting mutations, E1:K211E and E2:V264A. Further to the above two mutations, 11 of 15 CHIKV genomes from the Mombasa outbreak showed a novel triple mutation signature of E1:V80A, E1:T82I and E1:V84D. These novel mutations are estimated to have arisen in Mombasa by mid-2017 (2017.58, 95% HPD: 2017.23, 2017.84). The MRCA for the Mombasa outbreak genomes is estimated to have been present in early 2017 (2017.22, 95% HPD: 2016.68, 2017.63). Interestingly some of the earliest genomes from the Mombasa outbreak lacked the E1:V80A, E1:T82I and E1: V84D substitutions. Previous laboratory experiments have indicated that a substitution at position E1:80 in the CHIKV genome may lead to increased CHIKV transmissibility by *Ae. albopictus*. Genbank investigation of all available CHIKV genomes revealed that E1: V80A was not present; therefore, our data constitutes the first report of the E1:V80A mutation occurring in nature. To date, chikungunya outbreaks in the Northern and Western Hemispheres have occurred in *Ae. aegypti* inhabited tropical regions. Notwithstanding, it has been suggested that an *Ae. albopictus* adaptable ECSA or IOL strain could easily be introduced in these regions leading to a new wave of outbreaks. Our data on the recent Mombasa CHIKV outbreak has shown that a potential *Ae. albopictus* adapting mutation may be evolving within the East African region. It is even more worrisome that there exists potential for emergence of a CHIKV strain more adapted to efficient transmission by both *Ae. albopictus* and *Ae.aegypti* simultaneously. In view of the present data and history of chikungunya outbreaks, pandemic potential for such a strain is now a likely possibility in the future. Thus, continued surveillance of chikungunya backed by molecular

PRJNA655685. Sequences are deposited GenBank accession numbers: MT380146-MT380162.

**Funding:** This work was supported by the Armed Forces Health Surveillance Centre (AFHSC), Division of Global Emerging Infections Surveillance and Response System (GEIS) FY2019 ProMIS ID P0136_19_KY_12.01 (RS) and P0136_19_KY_12.05 (RS). The funders had no role in study design, data collection and analysis, decision to publish, or preparation of the manuscript.

**Competing interests:** The authors have declared that no competing interests exist.

epidemiologic capacity should be sustained to understand the evolving public health threat and inform prevention and control measures including the ongoing vaccine development efforts.

## Introduction

Chikungunya virus is a mosquito transmitted alphavirus that was first isolated during an outbreak of febrile illness in Tanzania in 1952 [1]. Since then, CHIKV has caused many outbreaks, widely distributed around the globe [2]. From mid-December 2017 to mid-May 2018, an outbreak of chikungunya fever occurred in the coastal county of Mombasa [3]. The Mombasa outbreak followed an earlier outbreak of chikungunya in Mandera county on the Kenyan border with Somalia in 2016 [4]. The Mandera outbreak occurred 12 years after the coastal Kenya outbreak which began in Mombasa city and Lamu Island concurrently [5]. That outbreak later spread to the Indian Ocean basin, South East Asia and Europe [6, 7]. During the Indian and Indian Ocean phases of the outbreak, convergent genome microevolution led to the E1-A226V amino acid substitution in the CHIKV glycoprotein. This resulted in a strain that was highly adapted for transmission by *Aedes albopictus* as seen in La Reunion and elsewhere [8, 9].

As the outbreak progressed more *Ae. albopictus* adapting mutations developed, however these mutations were observed to have no effect on transmissibility of the virus by *Ae. aegypti* [10–12]. In 2006 a mutation within the E1 protein, E1:K211E was detected for the first time in Kerala and Puducherry, India [13]. Subsequently in 2009 a second mutation within the E2 protein, E2:V264A, was also detected in several regions of India [13–15]. In the background of a wild type E2:226A, the two mutations increase chikungunya fitness for *Ae. aegypti* while having no effect on virus fitness for *Ae. albopictus*, [16]. The severity of these outbreaks were linked to the mutations in the envelope proteins E1 and E2 within the IOL strain [12, 17]. Mutations within the glycoprotein have been shown to increase CHIKV fitness by up to 100 fold for *Ae. Albopitus* transmission and up to 62 fold for *Ae. aegypti* transmission [16, 17].

The alphavirus glycoproteins E1, E2 and E3 are involved in virus interaction with host cells and therefore determine efficiency of disease transmission and the host immune response mechanism. Glycoprotein E1 mediates cell fusion [18], glycoprotein E2 is important for interaction with the Host receptors and glycoprotein E3 facilitates E1-p62 heterodimerization and prevents the exposure of E1 fusion loops prematurely [19, 20]. Thus, sequential adaptive mutations within the CHIKV genome and more so in the envelope protein may influence efficient virus circulation and persistence in endemic areas, and could further increase the risk of more severe, bigger, and expanded CHIKV epidemics [12]. Viral variant evolution may continue to generate strains with complicated pathogenicity as shown during the Indian and Indian Ocean outbreaks [8]. Additionally, there has been a gradual expansion in the range of areas inhabited by *Ae. albopictus* [21]. It follows then that any emerging *Ae albopictus* adapted chikungunya virus strains may be transmitted in these new areas [22–24].

In the present study, we characterized novel mutations in the E1 protein of viruses from the Mombasa 2017/2018 CHIKV outbreak. The objective was to compare genomes generated from this outbreak with those from the Mandera outbreak of 2016 and other IOL strains to determine the genome structure, molecular features and signatures unique to the Mombasa 2017/2018 outbreak genomes.

## Methods

### Ethics statement

The study was carried out on a protocol approved by the Walter Reed Army Institute of Research's Institutional Review Board (#2189.0003) and Kenya Medical Research Institute's Scientific and Ethics Review Unit (#3035) as an overarching protocol guiding investigation and reporting of arbovirus/hemorrhagic fever outbreaks in Kenya.

### Patient samples and sequencing

Following widespread incidence of febrile illness cases in Mombasa County in mid-December 2017, samples were collected through the Kenya Ministry of Health (KMoH). During the course of the outbreak spanning 5 months, samples were obtained from patients as per standard KMoH procedures. Briefly, a patient presenting with sudden onset of fever >38.5° C, headache, severe joint pains and/or muscle pains while residing in Mombasa County within the preceding 3–5 days was considered a case. RNA was extracted from 17 human serum samples using TRIzol reagent according to manufacturer's instructions (Invitrogen, Carlsbad, CA). All the samples were subjected to cDNA synthesis using superscript III (Invitrogen, Carlsbad, CA) followed by targeted amplification using a set of 8 overlapping PCR primer pairs (Table 1). Prior to library preparation, PCR amplicons for each of the primer pair were combined in equal volumes for each of the samples, followed by cleaning using Agencourt Ampure XP beads (Beckman Coulter, Beverly, MA, USA). The cleaned amplicons were quantified using the Qubit 3.0 dsDNA HS Assay Kit (Thermo-Fisher Scientific Inc., Wilmington, DE, USA). Library preparation was performed using Nextera XT DNA Sample Prep Kit (Illumina, San Diego, CA, USA) according to the manufacturer's instructions. Briefly, 1ng of the PCR amplicons was used as starting material. Nextera XT Index Kit (Illumina, San Diego, CA, USA) was used to uniquely barcode the samples. Libraries were normalized before pooling using standard library normalization process (Illumina, San Diego, CA, USA). Sequencing was performed on an Illumina Miseq platform using Miseq reagent kit V3 (Illumina, San Diego, CA, USA), generating 2 x 300 base paired-end reads. Raw sequence reads obtained from sequencing were initially inspected for quality using FASTQC [25]. Initial quality control was performed with prinseqLite [26]. Sequence reads from this study were submitted to the Sequence Read Archive (SRA) under reference PRJNA655685. Filtered reads were used as input in performing both de novo and reference-guided sequence assembly using SPAdes v3.10 [27] and NGS_Mapper v1.5 [28]. Polishing of the final consensus sequences and inspection for possible primer induced mutations were performed manually using BAM files generated by NGS_Mapper and visualized in the Integrative Genomics Viewer [29]. The 17 Sequences obtained from this study have been submitted to Genbank under accession numbers MT380146-MT380162.

**Table 1. Primers used for PCR.** Five combinations of the primers were used to generate amplicons for NGS sequencing.

| Sequence Name (Forward) | Sequence | Sequence Name (Reverse) | Sequence |
|---|---|---|---|
| 1CDSF_Kenya | 5'—ATGGATCCTGTGTACGTGGAC—3' | 3504R_Kenya | 5'—TAATAGGCCTGGAGGGAARATG—3' |
| 3332F_Kenya | 5'—TAATAGGCCTGGAGGGAARATG—3' | 7495R_Kenya | 5'—AGGACCGCCGTACAAAGTTAT—3' |
| 5065F_Kenya | 5'—TGCACAGGARGCGAGTACAATC—3' | 7495R_Kenya | 5'—AGGACCGCCGTACAAAGTTAT—3' |
| 7200F_Kenya | 5'—ACGATACTGTGACAGGAACAGCTTG—3' | 8403R_Kenya | 5'—GGAGCAGGGGAACGTGGTGTTCG—3' |
| 7200F_Kenya | 5'—ACGATACTGTGACAGGAACAGCTTG—3' | 10627R_Kenya | 5'—GTCTTCYCTCTCAGGCGTGCGACTTT—3' |

## Dataset generation and alignment

The generated sequences were combined with publicly available sequences from Virus Pathogen Resource database, including CHIKV sequences obtained during the outbreak in Mandera, northeastern Kenya in 2016. All the complete genome sequences (n = 991) were downloaded and combined with the sequences from this study. Each of these sequences was linked to the country of origin as well as the date of isolation. A manual literature search was done for sequences without relevant information; those that could not be assigned were excluded from further analysis. All the remaining sequences (n = 751) were then sorted according to the country of origin. Down-sampling of the sequences was performed on each of the countries from outside Africa using Cd-hit [30]. We used a similarity threshold of between 99.5 to 99.9 percent so as to end up with a relatively equal or near equal number of sequences while at the same time preserving the diversity of sequences from each of the individual countries. All the down-sampled sequences from the different countries were combined (n = 263).

Multiple sequence alignment was performed with MAFFT v7 [31, 32]. The aligned sequences were trimmed leaving the two CHIKV open reading frames only. Thus, the 5' and 3' untranslated regions as well as the non-coding intergenic region between the non-structural and structural proteins were trimmed off. Furthermore, the region corresponding to the 7-aa deletion within the NSP3 region that is found among the Western Hemisphere Chikungunya viruses was also removed.—The alignment was further checked for identical sequences using Cd-hit with -C equivalent to 1. Only one sequence was picked from each cluster of similar sequences.

The final sequence alignment (n = 254) was scanned for recombination using RDP4 software suite [33]. Recombinant sequences were determined if positive recombination signal, P = 0.05, was detected by all the three primary methods used (RDP, Genecov and Maxchi) and at least one of the secondary methods (Recscan and Siscan).

## Phylogenetic analysis

Phylogenetic analysis was performed using both the Maximum Likelihood, IQTREE [34], and the Bayesian (MrBayes) [35] Methods. We first tested the best nucleotide substitution model for our alignment using jModeltest2 [36]. Consequently, the best-fit substitution model was determined as General Time Reversible with Gamma distribution, GTR+G. This model was applied to both the ML and Bayesian phylogenetic analysis. Maximum Likelihood analysis was performed for 2000 bootstrap replications. Metropolis Coupled Markov Chain Monte Carlo method (MCMCMC) was applied for MrBayes, with 5 million generations performed in duplicate with 25% as burnin. The convergence of runs was determined by the average standard deviation in split frequencies of less than 0.01 as suggested by the software developers [35].

## Molecular clock analysis and phylogeography

Molecular clock analysis and phylogeography was performed using only the sequences that fell under the ECSA lineage from our initial phylogenetic analysis. To determine the appropriateness of our dataset in the estimation of temporal parameters, we first assessed the root-to-tip regression of our sequence data using TempEst [37]. Subsequently, analysis was performed using Bayesian Evolutionary tools by Sampling Trees (BEAST v1.10.4) [38]. Sequences were assigned to either of the following geographical regions; Africa, Americas, East Asia, Europe, Indian Ocean Is., SE Asia/Oceania and South Asia. The Kenyan sequences were given Kenya as their geographical trait location. The analysis was performed using a relaxed molecular clock model (uncorrelated log-normal) with GTR+G model of substitution as determined by

jModeltest2 [36]. A flexible non-parametric Bayesian Skyline model was used to estimate past population dynamics, and an asymmetric trait model was used to estimate ancestral states. Two independent runs of 500 million generations were performed and the output from the runs was combined after removal of 10% of the trees as burnin. To ensure adequate effective sample size (>200) was achieved, the logs from the analysis were assessed using Tracer [39]. The Maximum Clade Credibility Tree was visualized in Figtree.

### Selection pressure estimation

Selection pressure estimation was performed using both the ECSA lineage dataset used for BEAST analysis and the dataset used for ML analysis. Non-synonymous/synonymous (dN/dS) ratio substitution was estimated separately on the alignment corresponding to the non-structural and the structural region of the virus using the BEAST data as well as the Mombasa outbreak data. Site-specific selection was performed using FEL (p < 0.01) and FUBAR (prob. > 0.99), both applied within the Hyphy v2.5 package [40]. Episodic selection analysis was performed in the Datamonkey webserver [41] using Mixed Effect Method of Evolution (MEME; p < 0.01). Selection on a given site was considered present when all the three methods detected positive selection at that given site.

### Protein structure modelling

Mutations with potential functional significance that were unique to the Mombasa outbreak strain were mapped onto the CHIKV E1-E2 heterodimer (Protein Data Bank ID: 3N42) [42] using the Pymol Molecular Graphics System (The PyMOL Molecular Graphics System; DeLano Scientific, Palo Alto, CA).

## Results

### Origins of the chikungunya virus strain causing the Mombasa 2017/2018 outbreak

Seventeen (17) samples from the Mombasa 2017/2018 outbreak were sequenced. The sequenced samples covered the period between mid-December 2017 and mid-May 2018. Fifteen (15) of the samples produced near full CHIKV genomes, with two producing partial genomes with over 50% coverage. No recombination was detected in any of these genomes. Maximum likelihood and Bayesian phylogenies had a similar topology and showed that the Mombasa outbreak strain belonged to the CHIKV Indian Ocean sub-Lineage (IOL), within the ECSA lineage (Fig 1). All the Mombasa genomes clustered in a monophyletic clade with minimal diversity amongst them, as would be expected of samples from the same outbreak (Figs 1 and 2). The Mombasa outbreak cluster is closely related to the China strain of 2017 (Fig 2). The most recent common ancestor (MRCA) of the Mombasa 2017/2018 strain is estimated to have existed in early 2017 (2017.22, 95% HPD: 2016.68–2017.63). The Mombasa 2017/18 outbreak strain is estimated to have diverged from the 2017 China strain and the closely related Hong Kong 2016 strain in May 2016 (2016.43, 95% HPD: 2015.27–2017.11). The MRCA for the Mombasa, 2017–2018 strain and the Mandera 2016 strain is estimated to have existed late 2008 (2008.86, 95% HPD: 2007.34, 2010.90), (Fig 2). Whereas the Mandera outbreak originated from the India 2016 strain, the Mombasa outbreak originated from the India 2015 strain, whose MRCA is estimated to have existed in early 2011 (2011.27, 95% HPD: 2009.03, 2013.35) (Fig 2).

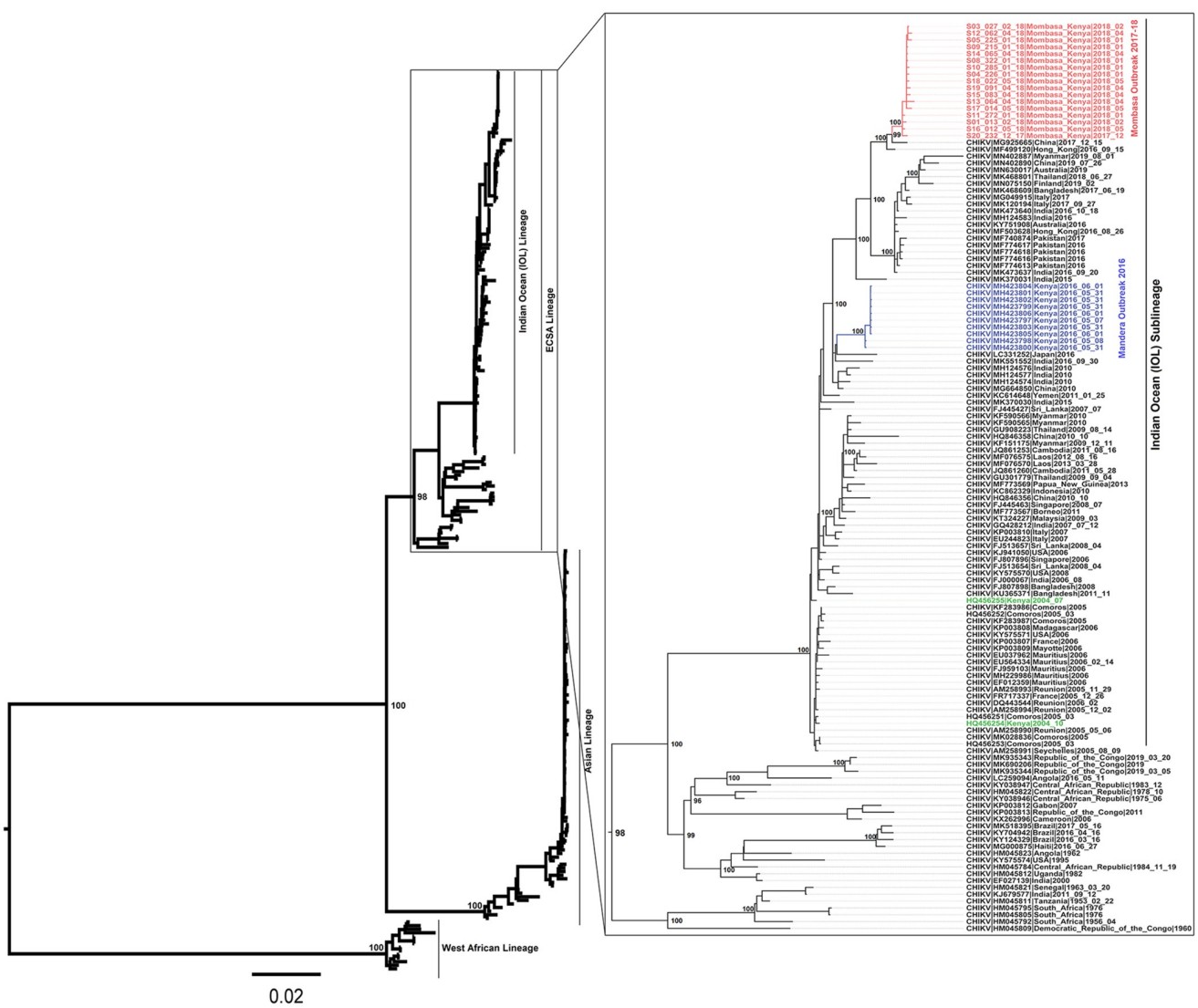

**Fig 1. Maximum likelihood phylogenetic tree based on the complete-coding region of the four CHIKV lineages.** The Mombasa 2017 outbreak genomes are highlighted in red and the Mandera outbreak strain is highlighted in blue. The Lamu/Mombasa 2004 strains are shown in green. The ECSA lineage with tip-labels is expanded.

## Novel mutations associated with increased fitness to *Ae. albopictus* emerged in Mombasa in mid-2017

Investigation of amino acids in Mombasa genomes revealed the presence of novel potential *Ae. albopictus* adapting glycoprotein mutations [43] in 11 of 15 CHIKV genomes. The E1: V80A, E1:T82I and E1:V84D mutations (Fig 2; in red) are herein reported for the first time in nature. This triple mutation signature seems to have emerged in Mombasa in mid- 2017 (2017.63, 95% HPD = 2017.33–2017.87). Four genomes from the outbreak did not have any of the three mutations (Fig 2; in purple). These Four are estimated to have existed within Mombasa early 2017 (2017.22, 95% HPD = 2016.68–2017.63). The other Mombasa unique substitution is E1:L136F. Another substitution observed in the present study is NSP4: R85G, which was present in the East Asia strain of 2015–2017 and maintained in the Mombasa 2017/2018

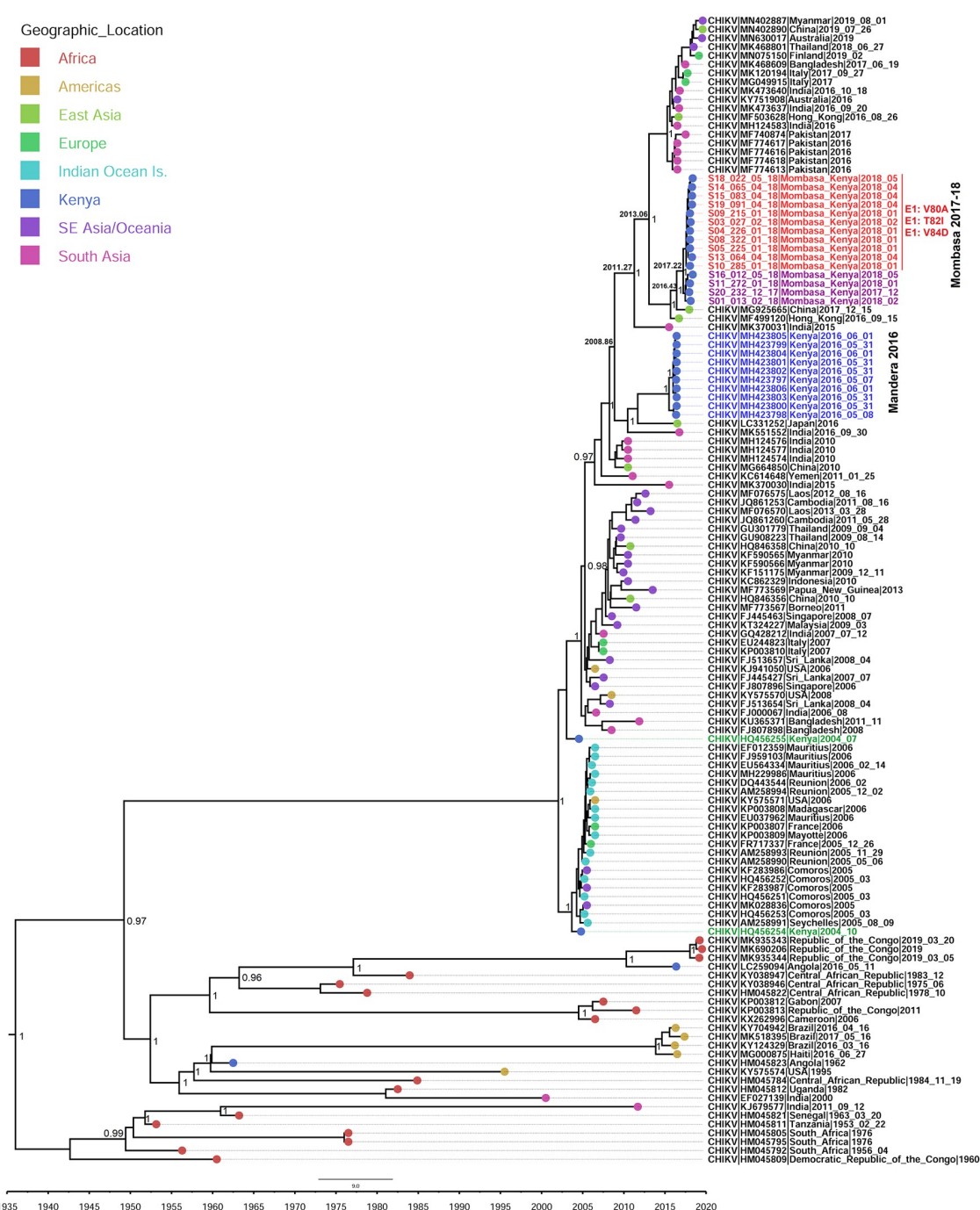

**Fig 2. Maximum clade credibility (MCC) tree generated using BEAST analysis.** Generated using 130 CHIKV sequences of the ECSA lineage. Mombasa genomes with E1:V80A, E1:T82I and E1:V84D mutations are shown in red, while those lacking these mutations are shown in pink.

genomes. The R85G Change is lacking in the Mandera 2016 strain. Other substitutions observed in the Mombasa/Hong Kong/China 2017 clade but absent in the Mandera 2016 strain include E2:M74I, E2: A76T and Capsid: N79S. These substitutions were also observed in genomes within the South East Asia 2012–2017 clade.

**Table 2. Site-specific selection pressure estimates based on two different datasets (BEAST = 247 genomes and ML = 130 genomes).**

| Datasets | | FEL (p < 0.01) | FUBAR (p < 0.01) | MEME prob. > 0.99 |
|---|---|---|---|---|
| ML dataset | Nonstructural protein | NSP1: 171 | NSP1: 171 | NSP1: 171, NSP2: 122, NSP2: 457, NSP3: 516, NSP4: 81, NSP4: 466, NSP4: 467, NSP4: 496 |
| | Structural protein | | | E2: 57, E2: 221, E1: 146 |
| BEAST dataset | Nonstructural protein | NSP1: 171 | NSP1: 171, NSP4: 467 | NSP1: 171, NSP2: 457, NSP3: 516, NSP4: 81, NSP4: 466, NSP4: 467 |
| | Structural protein | | | E1: 146 |

The results of the three different methods; FEL (p value < 0.01), FUBAR (prob. > 0.99) and MEME (p value < 0.01) are given.

## Positive selective pressure was acting on NSP1-171

The present study investigated the role of positive pressure on the appearance of the observed mutations in relation to virus adaptation. Whereas none of the new mutations observed were under selective pressure, significant positive selection was detected at position NSP1:R171 by all the three methods used, namely, FUBAR (probability> 0.95), FEL (P < 0.05), and MEME (P < 0.01) in both datasets. Additionally, for the present dataset positions E1: 210, E1: 211 and E2:264 did not show any evidence of positive selection (Table 2).

## The E1:V80A, E1:T82I and E1:V84D signature induced structural change within CHIKV E1-E2 molecular structure

The Mombasa unique molecular signature, the triple mutation of E1:V80A, E1:T82I and E1: V84D, caused a conformational change when compared to the wild-type (Fig 3B). Among the three mutations, it is only the E1:V84D substitution that introduced a negative charge.

## Discussion

Chikungunya glycoproteins have been shown to play a critical role in the emergence, transmission and spread of the virus. The E1:A226V substitution, which adapts CHIKV to *Ae. albopictus*, *is* credited with the IOL global outbreaks that occurred after the initial 2004 coastal Kenya outbreak. Beyond this initial *Ae. albopictus* adapting mutation, CHIKV has continued to undergo genome evolution, bringing about incremental fitness for both *Ae. albopictus* and *Ae. aegypti* [12, 14]. Coincidentally, the presence of the E1:A98T mutation in the Asian genotype that has recently circulated in the Americas is believed to constrain Chikungunya from infecting *Ae. albopictus* [11, 44]. Notwithstanding, it has been suggested that an *Ae. albopictus* adaptable ECSA or IOL strain could easily be introduced in these regions leading to a new wave of outbreaks [44, 45]. Even though no *Ae albopictus* populations have been detected in Kenya, this invasive species of mosquito has recently been detected in countries in West Africa and Central Africa [46, 47]. Phylogenetic analyses of the IOL have reported continued evolution of CHIKV since 2004 [48]. These analyses continue to track genome changes in relation to vector transmission. Our phylogenetic analysis adds to this growing body of data on the IOL (Figs 1 and 2), with the time-calibrated phylogeny showing a short period between the appearance of the Mandera and Mombasa strains (Fig 2).

The present study reports a novel CHIKV strain with a potential *Ae. albopictus* adapting mutation; V80A, evolving within the Kenyan Coast. It is instructive that the V80A substitution and its ability to adapt CHIKV to efficient transmission by *Ae. albopictus* was hitherto demonstrated within laboratory settings [10, 43, 44]. We suggest that two other mutations in the Mombasa strain, E1:T82I and E1:V84D, that are adjacent to V80A, might modulate the action

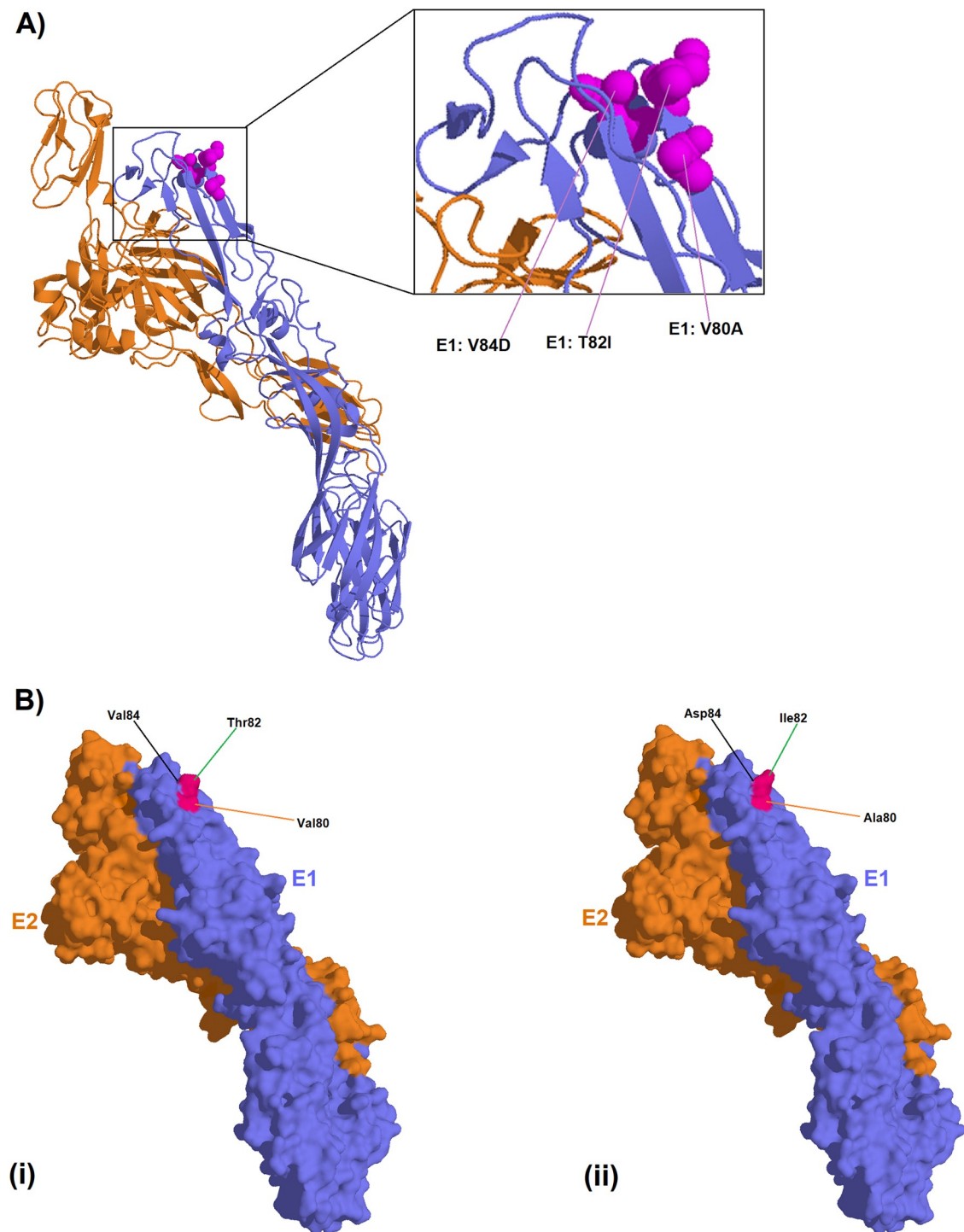

**Fig 3.** Protein figures showing; A) ribbon representation of CHIKV E1-E2 heterodimer (ID: 3N42) with the E1 monomer represented in blue and the E2 monomer represented in orange. Amino acid residues with potential functional significance are represented in spheres within the figure. B) Surface representation of the same E1-E2 heterodimer showing, i) wild type amino acids and ii) the reported mutations.

of the V80A substitution. Especially since, they also lie within the CHIKV fusion loop (Fig 3A). In fact, previous mutagenesis studies showed that a substitution at position E1:82 arises as second step to that of position 80 [43]. Thus, this may lend credence to the suggestion that the E1:T82I substitution in the Mombasa 2017 CHIKV strain potentially supports the E1:V80A action. Interestingly, a substitution at position 84 has not hitherto been demonstrated elsewhere, *in-vitro*, or otherwise. However, the protein models generated for genomes in this study reveal that together, the three mutations introduce conformational changes on the surface of the E1 protein (Fig 3B). The E1 glycoprotein residue at position 80 plays a central role in chikungunya infectivity and dissemination by modulating viral fusion and cholesterol dependence [43]. As such, whereas the E1-80 position is traditionally involved in CHIKV cholesterol-dependent entry, the E1-V80A variant abrogates the need for cholesterol use (34). Therefore, the Mombasa 2017 CHIKV strain may potentially be amenable to *Ae. Albopictus* transmission as observed in laboratory based studies elsewhere [10, 44]. If this should be the case, then the Mombasa 2017 CHIKV strain would be a potential public health concern, considering that the success of the IOL outbreaks during 2005–2007 were largely premised on the E1-V226A substitution, which caused the virus to have decreased need for cholesterol dependence for CHIKV fusion [49]. Furthermore, E1-V80A, was previously shown to be the most stable variant among all position 80 mutants [43]. Thus based on this published evidence, we suggest that, the observed micro-evolutionary genome changes in respect to position E1: 80, in the Mombasa 2017 strain might be stable and sustained in the long run if it were to spread to *Ae. albopictus* infested areas. Interestingly, the early samples from the Mombasa 2017 outbreak lacked the E1-V80A, E1-T82I, E1-V84D signature (Fig 1). Consistent with this pattern, the acquisition of the A226V substitution was observed later in the 2005 outbreak on Re-union Islands with all earlier samples showing the wild type 226A [8, 16]. Thus, we speculate that as with the acquisition of A226V on the reunion Islands, the observed triple mutation signature of E1-V80A, E1-T82I, E1-V84D may have solely been acquired in Mombasa. Therefore, continued surveillance will be necessary to decipher the extent of public health consequences of the E1-V80A, E1-T82I, E1-V84D signature, if any, beyond the Kenyan coast.

In addition to the novel amino acid substitutions associated with *Ae. albopictus* in this study, the E1: K211E and E2:V264A double mutation associated with *Ae. aegypti* adaptation [16], was observed. The E:1 K211E and E2:V264A double mutation was also observed in the 2016 Kenyan outbreak strain [4]. Whereas previous studies have shown strong positive selection for residues in the CHIKV E protein [4, 16], none seems to have been selected for the present study. On the contrary, the present study shows significant positive selection of position 171 in the NSP1 protein involved in virus replication (Table 2) pointing to potential evolution towards efficiency in viral replication. It will therefore be interesting to assess selection pressure on this strain in an *Ae. albopictus* environment. We suggest that more studies on the transmissibility of this strain be carried out in order to assess in-vivo capacity for transmission under field conditions.

The emergence of a CHIKV with ability for simultaneous efficient transmission by both *Ae. aegypti* and *Ae. albopictus* presents a worrisome public health scenario, especially for areas infested with *Ae.albopictus*. Notwithstanding, certain mutations may be confined to regions within which they were first reported. This was particularly observed for the L210Q substitution, which is thought to have been selected in Kerala India because of specific ecological conditions, since it has not been observed in any other strain [10, 50, 51]. Whether this might end up the same for the substitutions in the Mombasa IOL sub-lineage, can only be demonstrated through continued surveillance.

Finally, our phylogenetic analyses indicate that the Mombasa 2017/2018 and the Mandera 2016 outbreaks were not directly connected to each other. The trees show that the viruses from

the two outbreaks shared a common ancestor in late 2008 and were introduced into Kenya on two separate occasions. Both Mombasa and Mandera strains were most closely related to viruses circulating in Asia, suggesting standing routes of virus dissemination between Asia and Africa, and possibility of targeted prevention strategies once these routes are identified. Given that both strains also carried the E1:K211E and E2:V26A mutations further supports the notion of fixation and continuous spread of the E1: K211E and E2:V264A strains, resulting in additional outbreaks. Whereas there seemed to be no new defining genome evolutionary events with the Mandera strain, the Mombasa strain resulted in additional mutations with potential public health implications. With historical hindsight as far back as the 2004 CHIKV outbreak, continued surveillance of CHIKV in East Africa is advised.

## Conclusion

The current study has unveiled a novel CHIKV strain within the Mombasa 2017 outbreak. While providing an update on chikungunya evolution, the study points to the fact that the chikungunya virus may continue to present public health challenges on the global stage. The importance of this data should be seen in the light of providing health authorities with information to help generate policies that could mitigate against the potential spread of this potential emergent sub-lineage of the IOL, to new areas. This, more so when looked at in historical context whereby the subsequent spread of a novel lineage of the chikungunya virus that emerged in Coastal Kenya in 2004, caused debilitating disease in approximately 10 Million people with deaths in the 1000s being reported [6, 7, 52–54]. Furthermore, It has been suggested elsewhere [55] that East and Central Africa may have a large diversity of chikungunya strains that are co-circulating and are yet to be sampled. Continued surveillance and search for new strains will prepare health authorities with requisite information for control and vaccine development efforts.

## Acknowledgments

**Disclaimer**: The opinions and assertions contained herein are the private views of the authors and are not to be construed as official or reflecting the views of United States Army Medical Research Directorate-Africa, Kenya Medical Research Institute, Department of the Army, Department of Defense or the US Government. The investigators have adhered to the policies for the protection of human subjects as prescribed in AR 70–25.

## Author Contributions

**Conceptualization:** Fredrick Eyase, Rosemary Sang.

**Data curation:** Solomon Langat, Francis Mulwa, Albert Nyunja.

**Formal analysis:** Fredrick Eyase, Solomon Langat, Francis Mulwa, Albert Nyunja, James Mutisya, Victor Ofula.

**Funding acquisition:** Rosemary Sang.

**Methodology:** Solomon Langat, Irina Maljkovic Berry, Francis Mulwa, Albert Nyunja, James Mutisya, Samuel Owaka, Samson Limbaso, Victor Ofula, Hellen Koka, Edith Koskei, Joel Lutomiah.

**Project administration:** Fredrick Eyase, Rosemary Sang.

**Supervision:** Fredrick Eyase, Joel Lutomiah, Rosemary Sang.

**Validation:** Irina Maljkovic Berry, Albert Nyunja, James Mutisya, Samuel Owaka, Samson Limbaso, Victor Ofula.

**Writing – original draft:** Fredrick Eyase.

**Writing – review & editing:** Solomon Langat, Irina Maljkovic Berry, Francis Mulwa, Albert Nyunja, James Mutisya, Samuel Owaka, Samson Limbaso, Victor Ofula, Hellen Koka, Edith Koskei, Joel Lutomiah, Richard G. Jarman, Rosemary Sang.

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
