## [Decision Letter · Decision Letter 0]

16 Jul 2020

PONE-D-20-18596

Emergence of a novel chikungunya virus strain out of the Mombasa, Kenya 2017-2018 outbreak.

PLOS ONE

Dear Dr. Eyase,

Thank you for submitting your manuscript to PLOS ONE. After careful consideration, we feel that it has merit but does not fully meet PLOS ONE’s publication criteria as it currently stands. Therefore, we invite you to submit a revised version of the manuscript that addresses the points raised during the review process.

specifically, The 3 reviewers acknowledged the interest and quality of the results but asked for some improvement of the presentation and discussion. More importantly, more details about the NGS data have to be provided in order to give a stronger background as requested by the 1rst reviewer. As noted by the 2 ^rd^ reviewer, to adress the role of mutation in increasing CHIKV vector transmissibility, experimental proof have to be provided thus this part of the discussion should be balanced and is mainly (in the context of this article) suggestion for future work.

We look forward to receiving your revised manuscript.

Kind regards,

Pierre Roques, Ph.D.

Academic Editor

PLOS ONE

Journal Requirements:

2. We note that you are reporting an analysis of a microarray, next-generation sequencing, or deep sequencing data set. PLOS requires that authors comply with field-specific standards for preparation, recording, and deposition of data in repositories appropriate to their field. Please upload these data to a stable, public repository (such as ArrayExpress, Gene Expression Omnibus (GEO), DNA Data Bank of Japan (DDBJ), NCBI GenBank, NCBI Sequence Read Archive, or EMBL Nucleotide Sequence Database (ENA)). In your revised cover letter, please provide the relevant accession numbers that may be used to access these data. For a full list of recommended repositories, see http://journals.plos.org/plosone/s/data-availability#loc-omics or http://journals.plos.org/plosone/s/data-availability#loc-sequencing

3. Please include a copy of Table 2 which you refer to in your text (we note that currently there are two tables labelled as Table 1).

Reviewers' comments:

Reviewer's Responses to Questions

**Comments to the Author**

1. Is the manuscript technically sound, and do the data support the conclusions?

Reviewer #1: Partly

Reviewer #2: Yes

Reviewer #3: Yes

2. Has the statistical analysis been performed appropriately and rigorously? 

Reviewer #1: Yes

Reviewer #2: Yes

Reviewer #3: N/A

3. Have the authors made all data underlying the findings in their manuscript fully available?

Reviewer #1: Yes

Reviewer #2: Yes

Reviewer #3: Yes

4. Is the manuscript presented in an intelligible fashion and written in standard English?

Reviewer #1: Yes

Reviewer #2: Yes

Reviewer #3: Yes

5. Review Comments to the Author

Reviewer #1: While l agree with the authors that the emergence of CHIK in Ae. aegypti would potentially be a major issue for global public health, I have some fundamental issues with the strength of the argument they present. There are also some potentially important technical issues that need to be addressed.

Technical issues

1. the authors describe a targeted amplicon sequencing approach for CHIK, but while they present the primer sequences, they only indicate “5 combinations of these primers” but - not the specific combinations or predicted amplicons. this should be corrected.

2. Furthermore, they do not provide any data on the quality of sequencing, number of reads per library, chemistry used other than “NexteraXT” which isn’t detailed enough (2x150, 2x250? single end reads?). ALthough the genomes are deposited to GenBank, the raw sequencing reads are not available from SRA, thus preventing anyone from attempting to reproduce their results. This must be addressed.

3. the sequencing library approach they used is seriously flawed. PCR amplicons from separate single-plex reactions, which are then pooled in “equal volumes” and then cDNA purified and a library prep’d is a poorly thought out approach. There’s no ability to control for amplification efficiency across reactions, and the “equal volumes” of amplicons could have orders of magnitude differences in reads from each reaction.

4. the bioinformatics Pipeline is poorly described. De novo assembly of deep amplicon sequencing data is not a good approach unless you do substantial preprocessing of the raw reads to downsample your data. SPADES assembler inparticular can produce assembly graphs that make no sense simply by overloading the algorithm with large amounts of identicle reads, while maintaining a relatively low error rate. This is because sheer numbers of reads with errors will create incorrectly assembled contigs. Second - the “NGS_Mapper” tool they reference has... no reference... what is this? I’ve never heard of it, and there’s no citation for it. Was this a place holder the PI put in that the postdoc was supposed to “please fill in the mapper name” ? this is hopefully the error made here. In any case, even if they names the mapper - (read mapping the appropriate approach to take with known small RNA virus genomes... no assembly) - the pipeline must account for how the primers are trimmed and removed from the reads. Since they did not present any details on the amplicons, but indicated NexteraXT as the library prep chemistry, I’m left to assume that tagmentation was performed, thereby shredding the amplicons into semi random fragments (as we would expect). the issue though is that once this is done you must remove the primer sequences _after mapping_ your reference genomes - otherwise you will have significant and unwanted bias in your consensus sequences.

5. The comments made about protein structural changes need to be toned done. These are predictions ONLY, and are at best “potential” changes that “might” result in structural changes. Without structural / biochemistry studies - they can’t be sure.

6. The phylogenetic analysis of the paper is the strongest aspect of their study. They did a very thorough job on this and (correctly) used a GTR+g model for the ML tree. thumbs up on this.

Fundamental Issue

a/ lastly - the authors have discovered three new mutations from an outbreak in Mombasa. They use “potential(ly)” at least three times in the abstract - but claim this -potential- mutations are somehow connected with Ae. aegypti transmissibility. But... no lab work was done to show these mutations have anything to do with host fitness, virus life cycle, or anything to do with increased pathogenicity. At best - these are forensic markers for the outbreak they investigated - and while interesting that three mutations were unique to this outbreak in Mombasa - that’s as far as the authors should speculate.

I’m happy to discuss this review with the authors if they disagree.

Reviewer #2: The authors obtained 17 human serum samples positive to chikungunya virus (CHIKV) collected from the 2017-2018 Mombasa outbreak. They were able to sequence 14 full genomes and 3 partial genomes from these samples. The full genomes were placed in a phylogenetic context within the Indian Ocean sublineage (IOL), including the 2016 Manadera outbreak and the 2004 initial coastal Kenya outbreak. They didn’t find the Aedes albopictus adapting mutation E1:A226V but observed Aedes aegypti adapting mutations E1:K211E and E2:V264A previously detected in India. They also found mutations specific to the 2017-2018 Mombasa outbreak, E1:S210R, E1:V80A, E1:T82I and E1:V84D that could bring some increase of transmission in A. albopictus.

The analyses in the manuscript sound appropriate, figures are clear and the results support the conclusions. I have one major comment and few minors.

Major comment:

- The authors choose to mainly focus on describing mutations that could increase CHIKV vector transmissibility. I found that the manuscript lacks of results and discussion about the 2017-2018 Mombasa outbreak itself. The manuscript would benefit if the authors describe a bit more the phylogenetic patterns of the 2017-2018 Mombasa outbreak and discuss about its origin and spread.

Minor comments:

- A sequence assembly statistics table should be included in the manuscript to allow the readers to evaluate sequence and mutation robustness.

- The three partial genomes that are not included in the phylogenetic analyses could be added if they have at least 50% of sequence coverage

- A careful proofread of the manuscript is necessary to remove typos that are disseminated all along it

Reviewer #3: This study analyzed and compared 14 CHIKV complete genome sequences obtained from patients and identified a triple mutation signature (E1-V80A, E1-T82I, E1-V84D) and a double mutation (E1: K211E, E2:V264A) both associated with Aedes albopictus and Aedes aegypti adaptations in these CHIKV circulating in Mombasa, Kenyan coast. Conclusions state that a large diversity of CHIKV strains circulate in east Africa, which may emerge as a worldwide public health problem in the future, emphasizing necessity for constant surveillance and urgent vaccine development to protect people.

Title should reflect the mutations involved in mosquito adaptations (both species).

Please complete authors’ addresses

Please correct the word Alphavirus (genus) or alphavirus (in general when referring to any species from this genus). Alpha-virus is not correct (see ICTV website for virus taxonomy).

Backgroud lacks a little of “introduction” to the virus itself.

Please check for CHKIV incorrect abbreviation (correct for CHIKV).

Results: “a period spanning 5 months from mid-December 2017 to mid-May 2018.”

Two tables are assigned as “Table 1” (primer sets and site-specific selection pressure.

Phylogenetic trees are in low resolution and is not possible to read anything.

Please separate background and discussion paragraphs. Discussion does not consider phylogenetic trees, molecular clock, only the mutations (please discuss as well).

6. PLOS authors have the option to publish the peer review history of their article (what does this mean?). If published, this will include your full peer review and any attached files.

Reviewer #1: **Yes: **Jonathan L. Jacobs

Reviewer #2: No

Reviewer #3: No

---

## [Author Response · Author response to Decision Letter 0]

17 Sep 2020

Reviewer 1.

Technical issues:

1. The authors describe a targeted amplicon sequencing approach for CHIK, but while they present the primer sequences, they only indicate “5 combinations of these primers” but - not the specific combinations or predicted amplicons. This should be corrected. 

We concur with the reviewer, we have added specific combinations of primers and made a correction in the table of primers. 

2. Furthermore, they do not provide any data on the quality of sequencing, number of reads per library, chemistry used other than “NexteraXT” which isn’t detailed enough (2x150, 2x250? single end reads?). Although the genomes are deposited to GenBank, the raw sequencing reads are not available from SRA, thus preventing anyone from attempting to reproduce their results. This must be addressed.

We agree with the reviewer, we have corrected this section. We have also now deposited raw reads to SRA and updated the results section. The SRA reference is PRJNA655685.

3. The sequencing library approach they used is seriously flawed. PCR amplicons from separate single-plex reactions, which are then pooled in “equal volumes” and then cDNA purified and a library prep’d is a poorly thought out approach. There’s no ability to control for amplification efficiency across reactions, and the “equal volumes” of amplicons could have orders of magnitude differences in reads from each reaction.

We have used this approach before and seemed to have worked fine with publications at virus evolution and AJTMH. It seemed not to affect the eventual outcome of the sequencing run. Additionally, our study was not looking into minor variants such that we needed to equalize read coverage across the genome, we were only doing a consensus analysis. We appreciate the observation of the reviewer though and hope to consider it on our next project.

4. The bioinformatics Pipeline is poorly described. De novo assembly of deep amplicon sequencing data is not a good approach unless you do substantial preprocessing of the raw reads to down sample your data. SPADES assembler inparticular can produce assembly graphs that make no sense simply by overloading the algorithm with large amounts of identicle reads, while maintaining a relatively low error rate. This is because sheer numbers of reads with errors will create incorrectly assembled contigs. Second - the “NGS_Mapper” tool they reference has... no reference... what is this? I’ve never heard of it, and there’s no citation for it. Was this a place holder the PI put in that the postdoc was supposed to “please fill in the mapper name”? This is hopefully the error made here. In any case, even if they names the mapper - (read mapping the appropriate approach to take with known small RNA virus genomes... no assembly) - the pipeline must account for how the primers are trimmed and removed from the reads. Since they did not present any details on the amplicons, but indicated NexteraXT as the library prep chemistry, I’m left to assume that tagmentation was performed, thereby shredding the amplicons into semi random fragments (as we would expect). The issue though is that once this is done you must remove the primer sequences _after mapping_ your reference genomes - otherwise you will have significant and unwanted bias in your consensus sequences.

We agree that de novo assembly can introduce some errors, however we used both de novo and reference mapping to confirm the genomes. We agree and appreciate the reviewer on the primer issue and we have done manual curation, adding primer position to ensure that any primer induced mutations are removed, we have generated corrected consensus sequences and removed 1 mutation that was falling at position 210 of E1 and edited the manuscript accordingly. The corrected sequences have been resubmitted. No other mutation fell under the primer binding positions. We have also now referenced the NGS mapper which has been used in a number of projects before. We are grateful to the reviewer for pointing this out.

5. The comments made about protein structural changes need to be toned done. These are predictions ONLY, and are at best “potential” changes that “might” result in structural changes. Without structural / biochemistry studies - they can’t be sure.

We concur with the reviewer and we have rewritten the section on the protein structural changes.

6. The phylogenetic analysis of the paper is the strongest aspect of their study. They did a very thorough job on this and (correctly) used a GTR+g model for the ML tree. Thumbs up on this.

We appreciate the comments by the reviewer.

7. Fundamental Issue

 lastly - the authors have discovered three new mutations from an outbreak in Mombasa. They use “potential(ly)” at least three times in the abstract - but claim this -potential- mutations are somehow connected with Ae. aegypti transmissibility. But... no lab work was done to show these mutations have anything to do with host fitness, virus life cycle, or anything to do with increased pathogenicity. At best - these are forensic markers for the outbreak they investigated - and while interesting that three mutations were unique to this outbreak in Mombasa - that’s as far as the authors should speculate.

The mutations referred to as being connected with Ae. aegypti transmissibility are E1:K211E and E2:V264A, these two have previously been observed in many other studies and competence studies done elsewhere (referenced in the Manuscript) to show host fitness. These two mutations were observed in the Kenyan outbreak of 2016 (referenced in the Manuscript) and the present study. The New mutations observed are discussed in connection to their action on Ae. albopictus, one of the mutation E1:V80A has previously been characterized in a laboratory setting, our claim is that we have observed three new mutations for the first time in nature and that previous laboratory based experiments have demonstrated the ability of one of the mutations, namely E1:V80A to increase transmissibility of CHIKV by Ae. albopictus. These studies as referenced increased transmissibility by Ae. albopictus due to abrogation of the need to use cholesterol. Having clarified this, we concur with the reviewer and have toned down on the language around our claim.

I’m happy to discuss this review with the authors if they disagree

Reviewer #2: 

The authors obtained 17 human serum samples positive to chikungunya virus (CHIKV) collected from the 2017-2018 Mombasa outbreak. They were able to sequence 14 full genomes and 3 partial genomes from these samples. The full genomes were placed in a phylogenetic context within the Indian Ocean sublineage (IOL), including the 2016 Manadera outbreak and the 2004 initial coastal Kenya outbreak. They didn’t find the Aedes albopictus adapting mutation E1:A226V but observed Aedes aegypti adapting mutations E1:K211E and E2:V264A previously detected in India. They also found mutations specific to the 2017-2018 Mombasa outbreak, E1:S210R, E1:V80A, E1:T82I and E1:V84D that could bring some increase of transmission in A. albopictus.The analyses in the manuscript sound appropriate, figures are clear and the results support the conclusions. I have one major comment and few minors.

Major comment:

The authors choose to mainly focus on describing mutations that could increase CHIKV vector transmissibility. I found that the manuscript lacks of results and discussion about the 2017-2018 Mombasa outbreak itself. The manuscript would benefit if the authors describe a bit more the phylogenetic patterns of the 2017-2018 Mombasa outbreak and discuss about its origin and spread.

We have added information on the origin and spread into Mombasa of this outbreak strain of CHIKV. However, since there is no much differences among the Mombasa genomes and the outbreak took only 5-6 months, there is not much phylogenetic diversity among the genomes save for the a rising of the three Mombasa unique mutations. We have presented the arising of the unique mutations under the results section. We also report four genomes that seem not to have these mutations and indicate that these four are estimated to have existed within Mombasa in early 2017. Thus we show that the mutations potentially evolved within Mombasa. We have also indicated this in the discussion section. We have added the information on the origin of the Mombasa and Mandera strains as shown in the both ML and Bayesian phylogenies. As for the effect of the mutations, we have suggested further studies under field conditions.

Minor comments:

- A sequence assembly statistics table should be included in the manuscript to allow the readers to evaluate sequence and mutation robustness.

We have deposited the reads at RSA as suggested by reviewer 1 and believe the provided information together with the genomes would provide the information suggested by reviewer 2.

- The three partial genomes that are not included in the phylogenetic analyses could be added if they have at least 50% of sequence coverage

We concur with the reviewer, we have added the three genomes to the ML phylogeny as they had more 50% sequence coverage. We have also added one of the sequences to the Bayesian analysis as it is near complete.

- A careful proofread of the manuscript is necessary to remove typos that are disseminated all along it

We concur with the reviewer, and have read through the Manuscript to correct typological errors.

Reviewer #3: 

This study analyzed and compared 14 CHIKV complete genome sequences obtained from patients and identified a triple mutation signature (E1-V80A, E1-T82I, E1-V84D) and a double mutation (E1: K211E, E2:V264A) both associated with Aedes albopictus and Aedes aegypti adaptations in these CHIKV circulating in Mombasa, Kenyan coast. Conclusions state that a large diversity of CHIKV strains circulate in east Africa, which may emerge as a worldwide public health problem in the future, emphasizing necessity for constant surveillance and urgent vaccine development to protect people.

Title should reflect the mutations involved in mosquito adaptations (both species).

Please complete authors’ addresses

We concur with the reviewer and have incorporated the Novel V80A mutation in the title found in the present study.

Please correct the word Alphavirus (genus) or alphavirus (in general when referring to any species from this genus). Alpha-virus is not correct (see ICTV website for virus taxonomy).

We concur with the reviewer and have corrected to alphavirus as correctly pointed out

Backgroud lacks a little of “introduction” to the virus itself.

We concur with the reviewer and have added introductory information on the virus 

Please check for CHKIV incorrect abbreviation (correct for CHIKV).

We concur with the reviewer and have corrected to CHIKV as correctly pointed out by the reviewer

Results: “a period spanning 5 months from mid-December 2017 to mid-May 2018.”

The section has been changed to read “The sequenced samples covered the period between mid-December 2017 and mid-May 2018.”

Two tables are assigned as “Table 1” (primer sets and site-specific selection pressure.

The tables have now been reassigned to Table 1 and Table 2 respectively.

Phylogenetic trees are in low resolution and is not possible to read anything.

The Phylogenetic trees are provided as .tif images, if you click at the blue icon at the top right of the PDF image, you get clearer images

Please separate background and discussion paragraphs. 

Background and discussion paragraphs have been separated

Discussion does not consider phylogenetic trees, molecular clock, only the mutations (please discuss as well).

We concur with the reviewer and have added a section in the discussion on IOL Phylogeny including the Mandera and Mombasa outbreak strains

---

## [Decision Letter · Decision Letter 1]

21 Oct 2020

Emergence of a novel chikungunya virus strain bearing the  E1:V80A substitution, out of the Mombasa, Kenya 2017-2018 outbreak.

PONE-D-20-18596R1

Dear Dr. Eyase,

We’re pleased to inform you that your manuscript has been judged scientifically suitable for publication and will be formally accepted for publication once it meets all outstanding technical requirements.

Kind regards,

Pierre Roques, Ph.D.

Academic Editor

PLOS ONE

Additional Editor Comments (optional):

Reviewers' comments:

Reviewer's Responses to Questions

**Comments to the Author**

1. If the authors have adequately addressed your comments raised in a previous round of review and you feel that this manuscript is now acceptable for publication, you may indicate that here to bypass the “Comments to the Author” section, enter your conflict of interest statement in the “Confidential to Editor” section, and submit your "Accept" recommendation.

Reviewer #1: All comments have been addressed

Reviewer #2: All comments have been addressed

Reviewer #3: All comments have been addressed

2. Is the manuscript technically sound, and do the data support the conclusions?

Reviewer #1: Yes

Reviewer #2: Yes

Reviewer #3: Yes

3. Has the statistical analysis been performed appropriately and rigorously? 

Reviewer #1: Yes

Reviewer #2: Yes

Reviewer #3: Yes

4. Have the authors made all data underlying the findings in their manuscript fully available?

Reviewer #1: Yes

Reviewer #2: Yes

Reviewer #3: Yes

5. Is the manuscript presented in an intelligible fashion and written in standard English?

Reviewer #1: Yes

Reviewer #2: Yes

Reviewer #3: Yes

6. Review Comments to the Author

Reviewer #1: I want to thank the authors for addressing the concerns of the reviewers on this important manuscript.

Reviewer #2: (No Response)

Reviewer #3: This second version is a lot improved on writing than the latest one. I do understand that these mutations leading to vector adaptations, as already demonstrated in literature, are quite concerning since it was already reported for Zika virus, previous to the emergence of the virus in the Americas. For this reason, this article should be published shortly, representing a very important contribution for arbovirus surveillance. Few comments are described above in order to contribute with author´s review.

Abstract: could be reformulated in a more organized and direct approach, still need work to achieve the same quality of writing of the rest of the manuscript. Please indicate in which CHIKV genotype these triple mutations were identified in the abstract, try to make the text more complete of results. It is not clear in the abstract, however in the manuscript/tree is possible to understand that they refer to IOL strains. Abstract should reflect more the results/discussion.

7. PLOS authors have the option to publish the peer review history of their article (what does this mean?). If published, this will include your full peer review and any attached files.

Reviewer #1: **Yes: **Jonathan L Jacobs

Reviewer #2: No

Reviewer #3: No

---

## [Editor Report · Acceptance letter]

26 Oct 2020

PONE-D-20-18596R1 

Emergence of a novel Chikungunya virus strain bearing the E1:V80A substitution, out of the Mombasa, Kenya 2017-2018 outbreak. 

Dear Dr. Eyase:

I'm pleased to inform you that your manuscript has been deemed suitable for publication in PLOS ONE. Congratulations! Your manuscript is now with our production department. 

Kind regards, 

on behalf of

Dr. Pierre Roques 

Academic Editor

PLOS ONE